# Scipion-EM-ProDy: A Graphical Interface for the ProDy Python Package within the Scipion Workflow Engine Enabling Integration of Databases, Simulations and Cryo-Electron Microscopy Image Processing

**DOI:** 10.3390/ijms241814245

**Published:** 2023-09-18

**Authors:** James M. Krieger, Carlos Oscar S. Sorzano, Jose Maria Carazo

**Affiliations:** Biocomputing Unit, National Centre for Biotechnology (CNB CSIC), Campus Universidad Autónoma de Madrid, Darwin 3, Cantoblanco, 28049 Madrid, Spain

**Keywords:** global protein dynamics, software integration workflows, cryo-electron microscopy, normal mode analysis, elastic network models, ensemble analysis, principal component analysis, hybrid simulations

## Abstract

Macromolecular assemblies, such as protein complexes, undergo continuous structural dynamics, including global reconfigurations critical for their function. Two fast analytical methods are widely used to study these global dynamics, namely elastic network model normal mode analysis and principal component analysis of ensembles of structures. These approaches have found wide use in various computational studies, driving the development of complex pipelines in several software packages. One common theme has been conformational sampling through hybrid simulations incorporating all-atom molecular dynamics and global modes of motion. However, wide functionality is only available for experienced programmers with limited capabilities for other users. We have, therefore, integrated one popular and extensively developed software for such analyses, the ProDy Python application programming interface, into the Scipion workflow engine. This enables a wider range of users to access a complete range of macromolecular dynamics pipelines beyond the core functionalities available in its command-line applications and the normal mode wizard in VMD. The new protocols and pipelines can be further expanded and integrated into larger workflows, together with other software packages for cryo-electron microscopy image analysis and molecular simulations. We present the resulting plugin, Scipion-EM-ProDy, in detail, highlighting the rich functionality made available by its development.

## 1. Introduction

Macromolecular complexes exhibit considerable flexibility, which is essential for their diverse functions, including signal transduction, transport of small molecules and ions, enzymatic catalysis and mechanical work, and regulating interactions with other molecules [1]. They undergo a range of motions from small, local rearrangements to large-scale, collective/global conformational changes [1,2]. These motions result in rich ensembles and continuous conformational landscapes, which are starting to be resolved by a range of computational and experimental methods [3,4,5,6,7].

One very exciting area is cryo-electron microscopy (Cryo-EM) single particle analysis (SPA) where, thanks to an explosion of continuous heterogeneity analysis methods [6,8,9,10], it is now possible to start studying this flexibility from single-particle images. These state-of-the-art methods allow each particle image to be assigned some parameters in a multi-dimensional latent space that describe differences between them that are ideally related to their conformation and enable the visualisation of these spaces via dimensionality reduction, giving rise to landscapes where similar particles are usually in close proximity. They also provide tools for navigating the landscapes and clustering particles, and recovering maps from cluster centres and other points, using either a generative heterogeneous reconstruction autoencoder network [9], such as CryoDRGN (current stable version of which is 2.3.0) [11], or a deformation field [12,13,14].

### 1.1. The Scipion Flexibility Hub Solves Recent Solutions to Challenges with Cryo-EM Continuous Heterogeneity Analysis

Despite the great progress being made, these approaches have several intrinsic limitations (including experimental issues such as potential particle damage/denaturation and enrichment/depletion of particular orientations and conformations as a result of freezing [15] and the air water interface [16]), and it is very challenging to extract biological and physical meaning from the results of these methods as discussed in a recent review [10] (see Table 1, column 1). For example, it is now possible to generate large numbers (tens or hundreds) of maps fairly rapidly. Still, it is far from trivial to follow conformational changes between them and interpret the landscapes in either original or reduced latent coordinates. It is also difficult to compare and combine the results from different methods to come up with consistent conclusions.

Some approaches can help by analysing the resulting maps directly, such as structure mapping methods based on correlations or deformations [12,17], quantification of regions experiencing rotations and strains between maps [12,18], or visualisation of morphs between maps in ChimeraX using either standard linear interpolation or optimal transport [19,20]. Nevertheless, even after ruling out compositional heterogeneity with advanced tools, such as the voxel PCA and atomic structure-based occupancy analyses approaches in the Model-based Analysis of Volume Ensembles package (MAVEn v1.0) associated with CryoDRGN [21], the differences between maps can also be due to various reasons and be difficult to interpret in an atomistic manner. The latter could be overcome by flexibly fitting atomic models to these maps, and many methods are available to do this [22], but this is also challenging with so many maps. There are, however, several new multi-map fitting approaches aimed at addressing this, such as Phenix variability refinement (available in recent Phenix versions starting with the version dev-4799 [23].

This can be helped by continuous heterogeneity methods that use atomic models in the interpretation of images and maps via computational biophysics approaches, such as normal mode analysis (NMA) and molecular dynamics (MD) simulations (Table 1, column 2). Prominent examples include hybrid electron microscopy normal mode analysis (HEMNMA) [24], HEMNMA-3D [25], NMMD [26], and MDSPACE [27] within the Scipion plugin ContinuousFlex [13]. Another set of approaches use Bayesian inference and computational biophysics for comparing images to atomic structures, including BioEM [28], Cryo-EM Bayesian inference of free energy profiles (cryo-BIFE) [29], and Cryo-EM ensemble reweighting [30]. These methods all have limitations to the number of images or maps that can be analysed, but they produce very valuable insights and continue to be improved. For example, a recent deep-learning-accelerated extension called DeepHEMNMA [31] infers HEMNMA parameters for a large image set based on training with the results from a smaller one.

In this context, we recently created the Scipion Flexibility Hub [32], a framework for combining continuous flexibility methods into pipelines with shared interactive analysis tools connected to the rest of the Scipion workflow engine for Cryo-EM image analysis and structural biology (version 3.0) [33,34]. This framework provides common tools for several continuous heterogeneity methods, including the CryoDRGN deep reconstruction generative network (supporting version 1.0.0, 1.1.0, 2.1.0 and 2.3.0) [11], HEMNMA [24,35], MDSPACE [27], and related methods presented in ContinuousFlex v3.4.0 [13], and some approaches based on the Zernike3D deformation framework [12,36]. Altogether, this enables the creation and analysis of both maps and atomic models for representative points (e.g., cluster centres) in the conformational landscapes inferred from the particle images (see Table 1, column 2). The core of Scipion Flexibility Hub is a plugin called FlexUtils that provides the main common framework and the Zernike3D tools (parts 1 and 2 in Figure 1).

The atomic model creation is achieved using a fast and approximate flexible fitting approach based on Zernike3D available in Xmipp since v3.21.09 [12,32]. These rough atomic models have many issues as a result of the approximations made in the Zernike3D method, including a lack of any physics in the description and application of the deformations. Nevertheless, these models are a good first approximation of the conformational changes in the images and maps and provide a good starting point for further analyses and improvements, enabled by the new plugin Scipion-EM-ProDy that was briefly introduced with the Scipion Flexibility Hub [32], and is presented in more detail here together with the latest developments.

### 1.2. Scipion-EM-ProDy: A New Scipion Plugin for Better Interpretation and Simulation of Atomic Structures through Rapid Computational Biophysics

We introduced Scipion-EM-ProDy to enable a connection to computational biophysics approaches, which have been developed for extracting and interpreting ensembles and conformational landscapes and inferring functional mechanisms [37].

While all-atom simulations are very popular and under considerable development, resulting in a large number of dedicated software packages and force fields [38,39] and a plethora of enhanced sampling methods [40,41], they still have considerable limitations, including their computational cost and sensitivity to force fields inaccuracies [39], making it challenging to sample events on biologically relevant time scales such as microseconds and milliseconds, especially for large macromolecular assemblies studied by Cryo-EM.

Thus, coarse-grained models and methods are often preferable [42,43], and have been introduced in various software, including ProDy since version 1.0 [44,45], that we integrated into Scipion with our new Scipion-EM-ProDy plugin. Among these are two fast matrix decomposition methods, elastic network model (ENM) normal mode analysis (NMA) and principal component analysis (PCA) of structural ensembles [46], which are also available in a small number of other specialised software packages, such as Bio3D (version 2.0 and later) [47] and ElNemo [48] (part of the web server of the same name and used by ContinuousFlex as version 3.1), and web servers such as WEBnm@ [49] and Dynomics [50]. These methods rapidly yield robust modes of motions with clear biological relevance over an extensive collection of macromolecular systems [2,46], and have led to the development of many related methods, including hybrid simulations that combine NMA or PCA with very short (1 to 100 picoseconds [ps]) MD runs [41], including the recently developed NMMD [26] and ClustENMD [51].

ProDy is a popular Python package (now at version 2.4.1) with a long history of development and a wide user base as a result of its systematic and rich architecture as an application programming interface (API) and its focus on these fast analytic methods for analysing global protein dynamics [45]. This has enabled several applications to be built on top of it, including internal ProDy and evol applications for core tasks and the normal mode wizard (NMWiz) [52] within VMD (all versions since 1.9.2) [53], the collective MD (coMD) hybrid simulation package [54] that also uses VMD [53] and NAMD (including latest stable version 2.14) [55], and the NMA-assisted docking program LightDock (currently at version 0.9.4) [56].

ProDy provides several ENMs, including the Gaussian network model (GNM) [57], the anisotropic network model (ANM) [58] and several variants [59,60], and the rotating and translating blocks (RTB) model [61,62], as well as the option to provide others with different distance relationships interactively [63]. It also provides many tools, such as the deformation vector approach to pairwise conformational changes between two structures and a rich ensemble building and analysis toolkit based on sequence and structural alignment and PCA, which allow comparison between experimental and computational results, and the hybrid simulation methods ClustENM and ClustENMD [51,64].

The creation of Scipion-EM-ProDy enables these tools to be more readily available to the general structural biology community, enabling a greater connection between experiments and computational biophysics (see Table 1, column 3, and the general workflow in Figure 1), which is critical for obtaining a proper understanding of macromolecular dynamics and function [5]. Two key steps are ensemble analysis to generate a conformational landscape with interpretable reaction coordinates from PCA (part 3) and to use the ClustENM(D) simulation framework for refinement and conformational sampling (part 4). We illustrate its main functionalities and updates using a system that we have studied recently and extensively, namely the D614G SARS-CoV-2 spike [32,65].

## 2. Results

Here, we present an overview of the functioning of Scipion 3.0 and the Scipion-EM-ProDy plugin and then some new example pipelines and workflows, demonstrating the key functionalities provided by Scipion-EM-ProDy that help interpret and refine structures from continuous heterogeneity analyses and enable comparison of experimental and simulated structures. In particular, we propose roles for advanced ensemble analysis and fast and flexible ClustENM(D) simulation tools.

### 2.1. General Overview of Scipion and Scipion-EM-ProDy for Building Workflows for Computational Biophysics

The general idea behind Scipion is to integrate many software packages into complex workflows comprised of various interconnected protocols, whose outputs are the inputs to subsequent protocols [33]. These protocols are provided by plugins, including the Scipion core plugins [33], as well many others for external software, such as Xmipp v3.23.07 [66], Relion 3.1 and 4.0 [67], ContinuousFlex 3.4.0 [13] and ChimeraX 1.0 to 1.4 [19]. A key feature is interoperability by handling data in common objects and converting it between different formats required for different programs. It also streamlines the smooth download and installation of these various software packages in a way that handles and isolates their associated dependencies.

The plugin Scipion-EM-ProDy provides 18 new protocols with associated objects and viewers, as well as a menu bar suggesting possible workflows (see Figure 2a). Most of these belong to three main categories corresponding to the three main steps of a protein dynamics analysis workflow: 1. atomic structure operations (atom selection, modelling, and alignment), 2. core dynamics calculations (deformation vector analysis, normal mode analysis, ensemble analysis, Gaussian network model [GNM] analysis, and hybrid simulations), and 3. downstream dynamics analyses (mode comparison, projection onto landscapes and clustering). Each protocol is run through a form, such as that shown in Figure 2b, for an ensemble building step.

A first example workflow (Figure 2c) illustrates an ensemble analysis of the D614G spike. As above, there are three main steps coloured in yellow, blue and green for atomic structure handling (importing and selecting atoms), core dynamics calculations (ensemble construction and PCA) and analysis (landscape projection), respectively. In this case, the selection protocol is used to import three structures from a data set from Benton, Wrobel and co-workers [68] from the protein data bank (PDB) [69]. These represent the three main states of the spike with 3 receptor-binding domains (RBDs) down (PDB: 7BNM), 1 RBD up and 2 RBDs down (PDB: 7BNN), and 2 RBDs up and 1 RBD down (PDB: 7BNO), and the protocols make the trivial selection of the protein atoms in this case as an illustration. The ensemble building protocol takes all these three structures as input and aligns them in a default way using the Cα atoms. A more complicated example following our recent paper [65] is described later. The principal component analysis takes the resulting ensemble of three aligned structures as input. It calculates two components of variation, and the projection protocol projects the atomic coordinates from the ensemble onto the two components.

Many of these steps have associated viewers, such as the ProDy normal mode viewer connecting to NMWiz 1.0 [44,52] in VMD 1.9.3 [53], shown at the bottom left, and the projection viewer based on Matplotlib (any version) [70], shown at the top right, which yields the plot below it. They can also use viewers from other plugins if they use common objects, such as those from ContinuousFlex.

As shown in Figure 2d, the arrows in NMWiz illustrate the active component of motion selected at the top of the window, which can also be animated by clicking the “make” button in the animation row of the bottom part. In this case, principal component (PC) 1 is shown, which shows a concerted closing and opening of two of the RBDs. PC2 shows an anti-correlated opening and closing of these two RBDs, which separates the 1-up structure (PDB: 7BNN) from the others. The landscape from projecting the ensemble onto these two PCs (Figure 2e) shows these separations. It separates the three structures via PC1 along the x-axis, with the 2-up structure (PDB: 7BNO) and the 3-down structures (PDB: 7BNO) being at the extremes, and the 1-up structure (PDB: 7BNN) being near zero, as PC1 is not relevant for this structure. Instead, the y-axis corresponding to PC2 separates the 1-up structure (PDB: 7BNN) from the other two structures via the closing of the second RBD as the first RBD opens.

### 2.2. Ensemble Analysis via PCA Downstream of Flexibility Hub Enables Interpretation of Cryo-EM Conformational Landscapes

Rather than starting from database structures, ProDy can also analyse structures from Cryo-EM analyses, such as those from the Zernike3D flexible fitting above (see Figure 1). While the resulting atomic models usually have many errors, especially at the level of local details, the global conformational changes should be fairly reasonable.

Following an iterative superposition with the ensemble protocol to converge the mean structure, PCA can then find and clean up the dominant conformational changes, allowing them to be used as new, interpretable coordinates for new conformational landscapes via vector projection (Figure 1, part 3). As each coordinate axis corresponds to a fairly simple mode of motion that can be visualised in NMWiz, this landscape is much more interpretable than the landscapes generated from the continuous heterogeneity methods themselves.

In principle, it should also be possible to project the particles into the PCA landscape via some appropriate interpolation approach. This could be aided by methods such as the conversion between Zernike3D deformations and principal components by thin plate spline interpolation [32].

### 2.3. A New ClustENM(D) Protocol for Refining Atomic Models and Hybrid Simulations

One recent addition to ProDy is the ClustENM class for the hybrid simulation approach ClustENMD [51] and its predecessor ClustENM [64] using OpenMM [71] for energy minimisation and MD. This method first fixes the starting atomic structure using PDBFixer [71], performs energy minimisation and optionally some picoseconds of MD simulations on it, and then runs several generations of the following steps: (1) generate several new conformations from each starting structure using random linear combinations of ENM normal modes (NMs), (2) cluster the resulting structures, and (3) minimise the cluster representative structures and optionally run some picoseconds of MD simulations for each. The combined approach is often referred to as ClustENM(D), with the D in brackets referring to the option of including the MD in ClustENMD or just running energy minimisation in ClustENM. Scipion-EM-ProDy has a ClustENM(D) protocol that can take multiple structures as input and run a ClustENM(D) simulation on each of them.

This protocol provides all the various options available within ProDy, which are spread across three tabs of the form (see Section 4.2.2). Amongst these, there is an option for the number of generations, which can be set to zero to perform energy minimisation and optionally a short MD simulation on the starting structure(s). It also provides the ability to run ClustENM(D) simulations on multiple structures simultaneously and then analyse them together. These two functionalities could be used to refine low-quality structures, such as those from Zernike3D fast flexible fitting, as discussed above (Figure 1, part 4).

### 2.4. A Combined Workflow for Comparing Structures from Experiments and Simulations

It is often useful to contextualise new experimental data in the context of existing structures, such as those deposited in the PDB or those obtained from simulations. We demonstrate such a workflow here using the set of SARS-CoV-2 spike structures with the D614G mutation available in the PDB (as in our previous paper [65]) and a ClustENM simulation based on one of them (Figure 3a). Structures from continuous heterogeneity analysis could also be included using the workflows presented above (see Figure 1).

After importing the set of 24 D614G spike structures studied previously [65] (grey box), this workflow contains three main pipelines. The first two (left half of Figure 3a) are based purely on the analysis of these experimental structures using fast analytical methods, starting with their alignment into an ensemble (red box). The core one (in the green shaded box) is based on PCA, as in Figure 2, and the other analyses the structure-encoded dynamics of some particular structures from root-mean-square deviation (RMSD) clustering (left orange box) using ENM NMA to inform the third workflow (blue shaded box) where we use one of these structures in an intermediate state (PDB: 7KEC) for ClustENM simulations. Ultimately, this allows us to compare the landscapes from the experimental and simulated ensembles by projecting them onto the same set of PCs (bottom right orange box) as shown in Figure 3b. The structure selection is guided by the correlation cosine overlaps between the mode of motion vectors from NMA and PCA (green boxes; Figure 3c).

Ensemble analysis of existing experimental structures can be challenging as different research groups often use different conventions for labelling residues and chains in their structures. However, as ProDy has been extensively used for analysing diverse structures, its advanced ensemble construction and analysis tools provide many options for matching chains [45,72], which are also provided by the Scipion-EM-ProDy plugin (see Section 4.1.3, and Figure 4).

In order to compare against simulations, we selected one of these structures for several generations of ClustENM and projected both the experimental and simulated ensembles onto the first two PCs from the experimental ensemble (see Figure 3c). To have a good starting point with relevant flexibility, we picked a structure with one RBD up and one RBD in an intermediate (I) state (1-up/1-I) (PDB: 7KEC) [73]. To inform this decision, we performed an RMSD clustering on the ensemble using an UPGMA tree and a cutoff of 2.5 Å, yielding seven cluster representatives belonging to a range of states including 3-down, 1-I, 1-up, 1-up/1-I, and 2-up. We then ran NMA for the most distinct cluster representatives and compared the first five NMs to the PCs from the ensemble. In contrast to the 3-down state, where there were only considerable overlaps to PCs 4 and 5, the first NMs of the 1-up/1-I state showed overlaps > 0.40 for the first two PCs (see Figure 3d). A similar behaviour with slightly higher overlaps was seen for the 2-up structure (PDB: 7EB4), but we decided not to use that state as it is more different to the others.

Prior to ClustENM, we used the separate PDBFixer protocol to fix missing residues and atoms and then selected the core structure without the termini. We then ran the ClustENM simulation with six modes and five generations of 50 conformers, each having an average RMSD of 2 Å from the previous conformer, and used the maximum number of clusters option with values of 10, 20, 30, 40 and 50 for the five generations. All other parameters were left at their default values.

In order to compare the two ensembles, we also trimmed away flexible loops that were missing in the experimental ensemble (see Section 4.1.2.2 and Figure 3e) and sliced the PCA vectors accordingly. This allowed the experimental and simulated structures to be projected onto the same landscape (Figure 3d), showing that the simulation (blue) explores the region between 3-down and 1-up, but does not explore as wide a region as the experimental structures (orange) and does not approach the 2-up state.

## 3. Discussion

Cryo-electron microscopy has advanced enormously over the last decade and now has the capability to fulfil its promise as a single-particle structural biology technique with a host of new computational methods capitalising on the potential to learn continuous structural heterogeneity information from each of the particle images in a data set [6,9,10]. All these methods generate landscapes that distinguish these images by various criteria, including compositional and conformational differences, and can generate estimates of maps in different conformational states for different regions or particles in these landscapes, but it is often difficult to interpret these results. We introduced a new plugin for ProDy within the Scipion workflow engine to help with this as illustrated in our various example workflows. In particular, we show how principal component analysis of atomic structure ensembles including existing structures can help contextualise conformational landscapes and illustrate that the hybrid simulation method ClustENM(D) can be used to rapidly improve structures and sample conformational space.

## 4. Materials and Methods

We first review the underlying implementation of Scipion and Scipion-EM-ProDy, and then highlight some key protocols and their associated pipelines, focusing on those with the richest functionality, namely those for (1) pairwise alignment of atomic structures and ensemble construction from multiple atomic structures; (2) ANM NMA and PCA; (3) GNM analysis and domain decomposition; and (4) ClustENM(D) hybrid simulations. We also include examples of key parameters, including those used in the example cases presented in Section 2.1 (Figure 2) and Section 2.4 (Figure 3).

All analyses presented here are based on publicly available structures from the protein databank (PDB) [69] and use standard computational biophysics methods available in ProDy as previously described in detail [45,74]. Scipion-EM-ProDy was installed in development mode, providing the latest development versions of ProDy and the plugin, similar to release versions 2.4.1 and 3.3.0, respectively.

### 4.1. Integration of ProDy Pipelines into Scipion Workflows

The Scipion workflow engine is primarily a collection of Python packages made up of several core packages plus a large number of plugins, which install and run various software. All the plugins import classes and methods from the Scipion core to enable the running of protocols through a common framework with common objects [33].

All software was written in Python. Scipion-EM-ProDy is a new Scipion plugin following standard Scipion plugin conventions, and is available at https://github.com/scipion-em/scipion-em-prody (accessed on 6 August 2023), as briefly described in [32]. Some changes have also been made to the Scipion-EM core package found at https://github.com/scipion-em/scipion-em (accessed on 1 August 2023), including the addition of the new object SetOfPrincipalComponents. Collaborative software development for these was performed through branches on the same fork with pull requests into the main devel branch and later into the master branches, as is standard for Scipion.

ProDy has already existed for over a decade [45], and has been developed further using its standard conventions and existing Github repositories, including https://github.com/prody/ProDy (accessed on 6 August 2023) and https://github.com/jamesmkrieger/ProDy (accessed on 6 August 2023). Collaborative development follows a personal forks and branches approach, with pull requests to the main prody/ProDy master branch, as is generally the standard for ProDy.

Scipion-EM-ProDy downloads ProDy from GitHub in two different ways. Either it downloads the tar.gz archive for the latest compatible release (currently v2.4.1), or it clones the latest compatible development code. These are then both installed in the same way, via two commands for building the C/C++ extensions in place and installing ProDy with pip in editable development mode.

#### 4.1.1. Building upon ProDy Classes, Functions and Apps to Create Scipion Protocols and Workflows

The Scipion-EM-ProDy plugin ensures a smooth hand-over of data from Scipion to ProDy and back through the use of corresponding objects (see Table 2). For Scipion, there are various classes of objects that point to files or items within them (among others) through the use of SQLite tables [33]. These pointers have file name attributes that can be passed on to programs such as ProDy to read the data from those files directly into memory as their own objects.

For example, the common step of a parsing an atomic structure uses the Scipion AtomStruct object that points to a file containing an atomic structure (in either PDB or PDBx/mmCIF format) that ProDy parses with the function parsePDB into Atomic objects (such as AtomGroup or Selection) containing information about all the atoms in the structure, such as residue names and numbers and Cartesian coordinates. After some calculations based on this structure, the results from ProDy are written to new files and registered in Scipion as metadata and file pointers in the SQLite (version 3.38.3) tables. In the case of atom selection, a new PDB file containing the subset of atoms, which is registered as another AtomStruct.

The main way to use ProDy is via the Python application programming interface (API), a Python package with some C and C++ extensions, providing a rich set of classes and functions for programmers to incorporate into pipelines either interactively or in scripts. There are also command line applications, which handle many of the key steps such as fetching structures from the PDB, selecting atoms, running ANM NMA, and performing PCA on simple ensembles, and are also accessible through an even more limited graphical interface inside NMWiz. Scipion-EM-ProDy gains the benefits of both of these two components: the Python API is useful for steps requiring more flexibility and not included in the apps such as alignment and ensemble construction using structures with multiple chains, and using the apps provides better control for the simpler and more computationally expensive activities and makes it easier to terminate the processes.

The Scipion-EM-ProDy plugin has been developed hand-in-hand with further developments to ProDy, to make this as smooth a process as possible. This includes the addition of new ProDy functions writeScipionModes and parseScipionModes for writing and parsing normal modes and related objects in Scipion format, where each eigenvector is written to its own file within a particle folder with one line of three values per atom and a single SQLite file with entries pointing to each one provides additional metadata such as eigenvalue, collectivity and a score based on the two from HEMNMA [24].

The apps were also updated to include additional features that are useful for including them in workflows, such as a limit to number of processors used and the option to include Hessian matrices in the NPZ (NumPy [75] pickle with gzip compression) files that ProDy uses for use in later steps, such as vibrational subsystem analysis (see Section 4.2.1).

We next present the protocols in the four main categories, in which they are used in a pipeline, as shown in Figure 2a.

#### 4.1.2. Protocols for Atomic Structure Operations

The first step in a ProDy pipeline is to parse one atomic structure or multiple atomic structures and perform some operations on them, such as reconstructing biological molecular assemblies, fixing the structure to account for missing atoms, selecting atoms, and aligning structures to each other in pairs or ensembles. These steps can also be used later in a process to help with the analysis, as in Figure 3. The corresponding protocols can fetch files from the PDB, read files from their paths, or take pointers to Scipion objects as inputs. The outputs are one or more PDB files registered as AtomStruct or SetOfAtomStructs pointers as discussed below, and there is also a summary of the number of structures and atoms.

The Scipion protocol for importing sets of atomic structures is especially useful here, allowing the download of several structures into files specified in a SetOfAtomStructs object via its associated SQLite table. This was particularly useful for the 24 D614G spike structures using the following PDB ids: 6ZWV, 7BNM, 7BNN, 7BNO, 7KDI, 7KDJ, 7KE4, 7KE6, 7KE7, 7KE8, 7KE9, 7KEA, 7KEB, 7KEC, 7KRQ, 7KRR, 7KRS, 7EAZ, 7EB0, 7EB3, 7EB4, 7EB5, 7DX1, 7DX2.

##### 4.1.2.1. Reconstructing Biological Molecular Assemblies

In several cases, especially when structures come from X-ray crystallography, the structure stored in the files downloaded from the PDB does not correspond to the biologically relevant assemblies (of which there can sometimes be several), and some symmetry operations are needed to reconstruct them. ProDy handles this through some arguments in parsePDB and the plugin provides a protocol to do this. It writes a PDB file for each assembly found, and the Scipion output is a SetOfAtomStructs object pointing to them.

##### 4.1.2.2. Atom Selection

It is often helpful to focus on particular chains, residues or atom types to obtain more meaningful results or efficient calculations. For example, NMA and PCA are often performed with Cα atoms as the other atoms are not needed to describe global motions [74] and the global rearrangements of particular domains can be important for signalling as is the case with N-terminal Venus fly trap domains from various receptors [76].

The plugin uses the select app, creating a new file, and the output is a new AtomStruct. The form has a box for entering a ProDy selection string, which is similar to that used by VMD (all versions). The syntax is described extensively with examples in tutorials on the ProDy website and a link to the relevant tutorial page is provided under the help option on the form. The default selection is “protein and name CA or nucleic and name P C4’ C2” in line with ProDy itself, based on various previous studies.

For the combined workflow in Figure 3, the selection string for trimming the termini prior to ClustENM simulations was “protein and resnum 26 to 1147”. We used the following selection string to trim loops to match the ClustENM ensemble to the experimental one, based on an analysis of missing atoms in the experimental ensemble in PyMOL: “name CA and resnum 27 to 66 82 to 113 116 to 140 166 to 172 186 to 196 200 to 211 215 to 242 264 to 442 491 to 501 503 to 515 522 to 620 641 to 676 690 to 811 813 to 827 856 to 939 944 to 956 958 to 1006 1008 to 1146”. The new structures have 2745 Cα atoms as compared to the original 51,737 atoms from 3366 residues in the full ClustENM structures.

#### 4.1.3. Pairwise Alignment and Ensemble Construction

Many comparisons should be made following alignment of atomic structures, either in pairs or larger numbers. This requires appropriate matching of residues and chains prior to structural superposition. ProDy provides several options for each type of matching [45,72] and performs superpositions using the Kabsch algorithm with the matching points [77].

The Scipion-EM-ProDy plugin provides two protocols for this: one for pairwise alignment and one for ensemble building. As in the ProDy API (where the function buildPDBEnsemble provides a wrapper for alignChains), these tasks and their protocol forms are very similar, and we mostly present the ensemble building one here (see Figure 4).

The pairwise alignment protocol has two fields for inputs, which select Scipion pointers for the mobile and target structures. The ensemble building one has the option to either take inputs from existing structures inside Scipion (which can be any combination of individual structures and sets of them) or to search for new ones using the DALI web server [78]. Currently, the DALI option cannot handle multiple chains as in ProDy itself, but single chain analyses can also be very informative and DALI alignments are pretty good, as shown in our signature dynamics (SignDy) paper [72].

For residue alignment, there are options to use Biopython pairwise sequence alignment [79] or combinatorial extension (CE) structural alignment [80] or to try both. There is also an option to use the mappings from DALI [78]. For chain matching, the options are (1) select same chain ID; (2) select same chain position; (3) use an automated matching scheme that optimises various features including sequence ID and alignment coverage for each chain and overall RMSD; and (4) manually input a custom dictionary of corresponding chain orders.

There are several cutoff options for accepting or rejecting alignments, which could rule out alignments from particular methods and favour the outputs of others or exclude particular structures from the ensemble altogether. In most cases, these are sequence identity and coverage for each chain and an RMSD for the whole structure. When using DALI, additional options are available, such as the DALI Zscore (see Figure 4b). These comparisons are all relative to the reference structure, which can be selected using an index from the provided structures or as another structure (see Figure 4c). There is also an option for whether to include dummy atoms when the reference structure has atoms that are not found in other structures or to trim them away, depending on an occupancy cutoff and a selection string, which allows the selection of particular chains, residues or atoms.

Figure 4d shows the custom chain matching implementation, which is based on two wizards (triggered by magic wand icons). The use of either wizard first results in the custom chain matching dictionary is populated automatically. Clicking the first wizard icon (circled in red) allows the dictionary to be updated by filling in the first two boxes with red labels and arrows. Values can also be recovered for checking or updating the matching order using the second wizard (circled in black), as indicated by the black arrows. The final dictionary used for the the D614G analysis is shown in Figure 4e, based on our previous study [65].

In this case, structures largely followed either one of two conventions from the first wildtype (WT) structures, with the first chain with an RBD up conformation as chain A and the other chains labelled B and C in an anticlockwise direction looking down on the RBDs (PDB: 6VSB) [81], or with the first chain with an RBD up conformation as chain B and the other chains labelled B and C in a clockwise direction (PDB: 6VYB) [82], creating chain orders ABC and BCA (see supplementary Table F in [65] and Figure 4c). There was also one structure from intact virions, which was determined around the same time as these two WT structures, and has a different arrangement again, resulting in a chain order BAC [83]. Additionally, there was one paper with two structures with two RBDs up with different chain IDs that needed to be made to match (PDB: 7EB4, 7EB5) [84]. We, therefore, used this approach again here in the ensemble building step (see Figure 4b,c).

The ensemble building protocol uses buildPDBEnsemble, which aligns each structure to the reference and then iteratively superposes the matching atoms to the average structure until the average converges, as this is necessary for PCA. The outputs (shown in Figure 4f) are (1) a SetOfAtomStructs object pointing to a number of PDB files with aligned structures based on the starting structures; (2) a ProDyNpzEnsemble object, which points to entries in ProDy’s specialised NPZ file for the ensemble; and (3) a sequence alignment where each row is a structure from the ensemble. The summary gives the number of structures and atoms in the ensemble, which in this case includes dummy atoms.

#### 4.1.4. Protocols for Calculating Global Modes of Motion

ProDy uses matrix decomposition methods to calculate global modes of motion, and these all work in similar ways, but with different matrices. Each one has a class based on the NMABase parent class with methods for setting coordinates, building the relevant matrices and calculating the modes from eigenvalue decomposition. This is handled by an internal utilities function solveEig, which calls relevant methods from Scipy (any version) [85] if available, or otherwise from NumPy versions ≥1.10 and <1.24 [75]. These classes are initialised and used in the corresponding apps, which are used the Scipion-EM-ProDy plugin. We briefly describe these here, but refer the reader to previous reviews (such as [41,74,86]) for more details. All these protocols produce NPZ files, which include the matrices and can be read back into ProDy, as well as NMD files for visualisation in NMWiz and sets of files in Scipion format with corresponding SetOfNormalModes and SetOfPrincipalComponents pointers. These pointers also have associated AtomStruct pointers to keep track of the starting structures.

The protocol output also includes animations generated using ProDy’s traverseMode function and corresponding VMD scripts and motion statistics for visualisation with the modes viewer from ContinuousFlex. All these forms look very similar too, with one tab for the main parameters and another for controlling the animations, e.g., setting the motion size in RMSD or the direction. We also provide a new ProDy mode viewer, which opens VMD and NMWiz to analyse the results. It can also read data from other SetOfNormalModes objects and write out new NMD files if they are not present.

##### 4.1.4.1. Deformation Vector Analysis

As well as calculating modes of motion from matrix decomposition, there is also an option of calculating a simple 3N-dimensional deformation vector q between two structures, describing the conformational change between them by subtracting the positions of the *N* corresponding atoms after alignment and superposition. This is equivalent to a morph, and is useful for both visualisation and comparison to the results from the other methods.

##### 4.1.4.2. Principal Component Analysis

PCA is based on decomposition of the 3N×3N covariance matrix of atomic positions from an ensemble of structures. This matrix is the average of the dot products $q\cdot q^{⊺}$ of the deformation vectors between each of the structures and the converged average structure. PCA decompositions finds the components of this covariance matrix that best explain the variance, which are usually interpretable conformational changes similar to those observed in morphs and more complicated methods. The input to this protocol is a pointer to an aligned ensemble as either a SetOfAtomStructs (which can also come from ContinuousFlex or FlexUtils) or a ProDyNpzEnsemble object. It also creates a fractional variance plot, which is saved as an image.

##### 4.1.4.3. Normal Mode Analysis Protocols

NMA is based on decomposition of the 3N×3N Hessian matrix of second derivatives of the interatomic interaction potential energy. This matrix can be built using various potential functions, but the most common and efficient ones are elastic network models (ENMs). Protocols for two common ENMs are provided here, namely the anisotropic network model (ANM) [58,87,88] and the rotating translating blocks (RTB) model [61,62]. Both of these are also provided by ContinuousFlex, and we have confirmed that it is possible to obtain very similar results, but ProDy is more flexible about options and can be more efficient in some cases. For the spike structures, we used the ANM with default parameters, including a cutoff of 15 Å and skipping zero eigenvalue modes. It is also possible to provide spring constant (gamma) and cutoff parameters as text to call other ENMs, such as GammaED and “2.9 * math.log(214) − 2.9” (replacing 214 for the number of residues in the protein used) for the essential dynamics (ED)-refined ENM [60], or GammaStructureBased for a version that takes into account secondary structure [59].

ContinuousFlex automatically chooses the model based on the input structure, and uses the RTB for atomic structures and the regular ANM for pseudoatoms [35], whereas ProDy can use either model for either situation. ProDy also does not provide an automatic relative cutoff option like ContinuousFlex, but it does provide suggestions on potential cutoff distances based on previous studies, such as 15 or 18 Å for Cα atoms [88], 5 to 7 Å for all atoms [87], or longer distances for fewer atoms, depending on the level of coarse-graining [89]. ProDy also has additional options, such as optimising blocks in RTB by splitting merging based on distance and minimum and maximum size or selecting them based on secondary structures. This overcomes some problems we have come across from having blocks with small numbers of residues or residues at opposite ends of loops with missing residues.

### 4.2. Gaussian Network Model Analysis and Domain Decomposition

The GNM is a related elastic network model from statistical mechanics of polymer networks [57], which has an N×N Kirchoff matrix and thus does not provide three-dimensional information on movements. It is based on the assumption of isotropic Gaussian fluctuations of atom positions and distance vectors, and incorporates more realistic constraints that result in better agreement with a wide range of experimental data [46]. This approach can be very useful for predicting mean square fluctuations and cross-correlations, and identifying critical regions such as hinge sites [46]. Two types of cross-correlation matrices are returned by the protocol: a raw covariance matrix and a normalised orientation cross-correlation matrix.

Scipion-EM-ProDy has a dedicated GNM viewer (see Figure 5a), which provides options for viewing mean square fluctuations (MSFs), root mean square fluctuations (RMSFs), and covariance and orientational cross-correlation matrices, estimated from either the full set of GNM modes or any individual mode or range of modes, as well as a per-atom maximum distance profile like that from ContinuousFlex and the GNM mode shape, which shows regions moving in opposite directions (Figure 5b), and to open NMWiz with structures coloured by GNM modes (Figure 5c).

GNM analysis can also be used in various pipelines, similarly to NMA and PCA, as shown in Figure 5d for an example using adenylate kinase (PDB: 4AKE, chain A). One useful application of GNM is dynamical domain decomposition, based on a spectral clustering using GNM modes [63,90], which is also provided in a protocol in Scipion-EM-ProDy. The output of this protocol is a PDB file with the b-factor column replaced by dynamical domain label and a VMD script to show a the structure as beads coloured by b-factors. There is then a viewer that displays this coloured structure (Figure 5e).

#### 4.2.1. Protocols for Downstream Analysis

Scipion-EM-ProDy also provides a number of protocols to assist with downstream analysis, including editing the modes to have similar numbers of atoms to others, comparing modes from different calculations, and projecting ensembles onto modes to visualise conformational landscapes.

##### 4.2.1.1. Mode Editing

It is often useful to apply slice operations to the eigenvectors from a set of modes, to ensure they have the same number of atoms as other modes or structures or ensembles, as we did for the comparison between experimental and simulated structures. ProDy also has an extend operation that allows eigenvectors calculated for Cα atoms to be copied to all atoms in the same residues, and the slice operation can be applied to do the opposite (see the light orange boxes in Figure 5d).

Another useful mode editing method is a Hessian reduction method, also known as subsystem-environment analysis or vibrational subsystem analysis, which calculates an effective Hessian for part of the system taking into account the effects of the rest of the structure as its environment [91,92,93]. This method is useful for assessing the impact on a particular domain from its interactions with the rest of the structure [93], as well as for accounting for the effect of a membrane [94,95].

These three mode editing methods have been incorporated as options in a single protocol, along with additional methods for thin-plate spline interpolation, which has been shown to be useful for extending coarse-grained modes based on pseudoatoms to atoms [96], that we added into ProDy as well. This protocol requires 3D modes, and cannot be used with GNM modes.

These protocols take a SetOfNormalModes or SetOfPrincipalComponents object as input, along with a new AtomStruct containing the new nodes, and compares them to find corresponding atoms using the same chain IDs and the default residue alignment, which first tries matching residue number then Biopython sequence alignment followed by CE structural alignment. The output is a new SetOfNormalModes or SetOfPrincipalComponents with an associated AtomStruct corresponding to the new nodes. Thus, these methods are currently limited to comparing modes based on atomic structures, but could be extended to pseudoatoms with methods such as nearest neighbours [63].

##### 4.2.1.2.Mode Comparison

Modes of motion can be compared in various ways [72,97], many of which are provided by ProDy. Scipion-EM-ProDy currently provides the correlation cosine overlap (also called inner products or projection products) [98], the root-weighted square inner product [99] and the covariance overlap [100]. The resulting matrix is written to a text file using ProDy’s writeArray function, and is registered in Scipion using the generic EMFile pointer. There is also an option to match and reorder modes based on linear assignment [72], which returns an additional file with mode matches and another SetOfNormalModes with the old IDs stored in the metadata. It is also possible to only calculate the diagonal values of the comparison matrix, equivalent to comparing the matching modes only. Lastly, there is an option to normalise overlaps or retain the raw projection products.

This protocol has an associated viewer (see Figure 5f) that can show the whole matrix as a heatmap (see Figure 3c) or particular rows (or the diagonal) using bar graphs, along with the cumulative overlaps as the square root of the sum of the squared overlaps if desired (see Figure 5g).

##### 4.2.1.3.Landscape Projection

The projection of an ensemble of structures onto a set of modes is a simple operation based on the dot product of the mode vectors and the deformation vectors from the average structures, which can be normalised or scaled to represent RMSDs. This can be shown as a set of points in a 2D or 3D scatter plot for two or three modes (see Figure 3b), or as a line for a single mode with the y-axis representing the projection and the x-axis being the conformation index. Alternatively, these can be approximated with a kernel density estimate for two modes or a histogram for one mode.

#### 4.2.2. ClustENM(D) Hybrid Simulations

The ClustENM class in ProDy enables ClustENM and ClustENMD hybrid simulations using OpenMM. Scipion-EM-ProDy has a protocol for running these simulations using this class, and the many parameters are spread across different tabs of the protocol form: one for general parameters, one for normal mode parameters and one for all-atom minimisation and MD simulation parameters. It has the option to take multiple structures as inputs and run an independent simulation for each one, making it useful for refining a set of structures from continuous heterogeneity analysis (see Figure 1).

#### 4.2.3. Protocols for Imports

Additionally, Scipion-EM-ProDy has two protocols for importing modes or ensembles calculated outside that Scipion project. The ensemble import protocol can also be used for trimming ensembles, as done in the pipeline in Figure 3a.

## Figures and Tables

**Figure 1 ijms-24-14245-f001:**
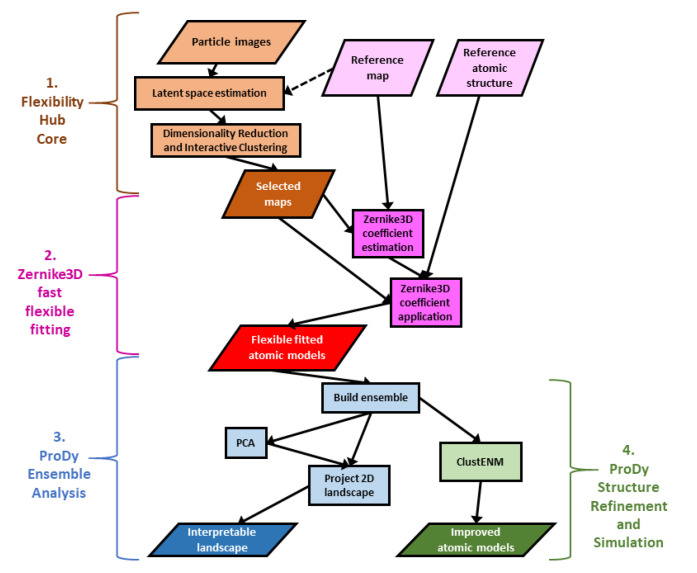
Illustrative workflow using Flexibility Hub and Scipion-EM-ProDy to solve challenges with interpreting landscapes from continuous heterogeneity analysis of single particle images and obtaining reasonable structures. There are 4 main parts. (**1**) A general Flexibility Hub starts from a set of particles from traditional SPA; (**2**) A fast, flexible fitting pipeline based on Zernike3D creates several structures; (**3**) ProDy ensemble analysis generates a conformational landscape; and (**4**) The ProDy ClustENM (D) protocol can be used for improving atomic structures.

**Figure 2 ijms-24-14245-f002:**
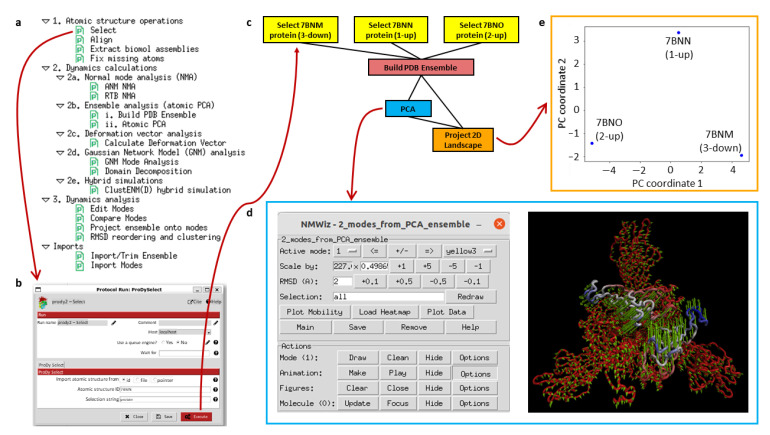
Overview of the Scipion-EM-ProDy plugin, with red arrows showing consecutive steps following user interaction. (**a**) List of protocols available within the plugin as shown on the left-hand side of a Scipion window. These are divided into categories with an order indicating how they should be used in pipelines and workflows. Double-clicking on one of these protocols yields a form as shown in panel (**b**). (**b**) An example protocol form is shown for atom selection. (**c**) Execution of protocols creates a workflow of inter-connected boxes, as shown. The workflow here illustrates importing of three D614G spike structures with selection of protein atoms (yellow) and ensemble construction and analysis via PCA (blue) and landscape projection (orange). (**d**) PCA results are shown with the ProDy mode viewer that opens up VMD and NMWiz (blue outlined box). The control window for NMWiz (left) has many options for visualising the structure with arrows and creating movies in the VMD structure viewer window (right). The average structure is coloured by the relative size of motion from most rigid in red to most mobile in blue. The green arrows show the first principal component where the two subunits slide past each other to close together (direction shown by arrows; towards 3-down) and open together (opposite direction; towards 2-up). (**e**) A projection plot from Matplotlib (orange outlined box) shows the conformational landscape from the ensemble in the space of the first two PCs, with each point being a structure.

**Figure 3 ijms-24-14245-f003:**
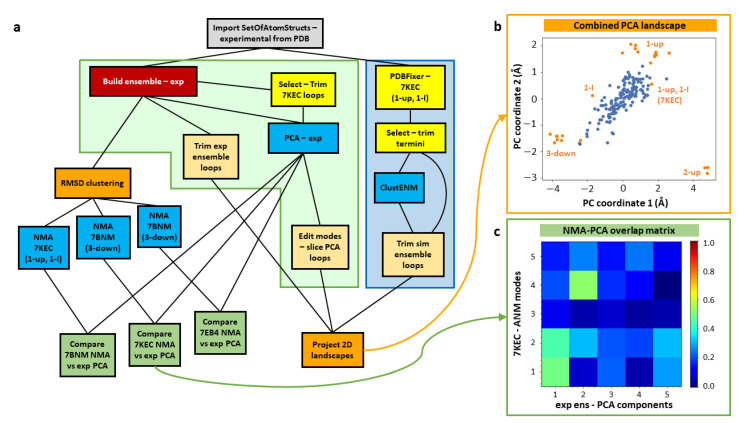
Combined workflow for comparing existing experimental structures and simulations. (**a**) The workflow contains two main parts: experimental structure ensemble analysis (green shading, left) and ClustENM simulations (blue shading, right). The experimental structures are aligned into an ensemble (red box) and analysed by PCA (central blue box) and RMSD clustering (left orange box). Some representative clusters are also used for NMA (3 blue boxes outside green shading) to confirm the best structure for simulations via comparison of NMs and PCs (following green boxes). Some atomic structure tools (yellow) were also used including as a first step to prepare the structure for simulations. Coloured arrows connect to other panels showing their outputs. (**b**) A combined landscape is shown projecting the experimental ensemble in orange and the simulated one in blue in the space of the first two principal components from the experimental ensemble (axis values are in units of RMSD in Å). The main conformations are labelled, including the simulation starting structure (PDB: 7KEC). (**c**) An overlap matrix is shown comparing the first five non-zero ANM normal modes of the 1-up, 1-I structure (PDB: 7KEC) to the first five principal components of the ensemble with low overlaps in blue and high overlaps in green, orange and red.

**Figure 4 ijms-24-14245-f004:**
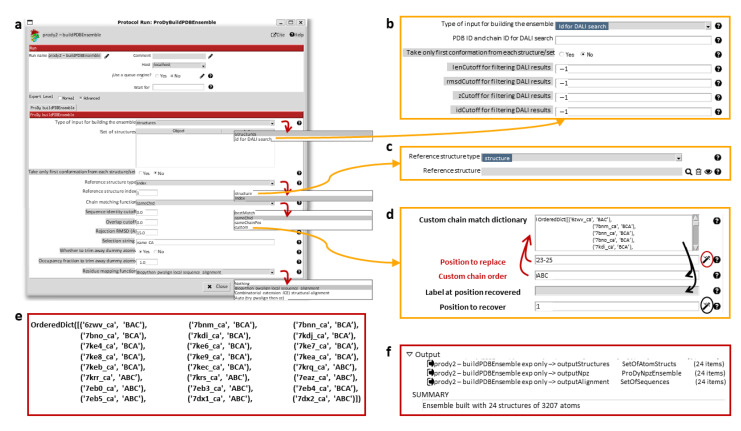
Technical details of the ensemble building form and its usage. (**a**) The ensemble building form is shown with each of the drop-down menus expanded to show the options. The alternative parts of the form that result from selecting particular options are shown in yellow boxes connected to arrows selecting these options. (**b**) Selecting the DALI option for input structures adds new filtering options for the results from the web server. (**c**) Selecting to use another structure as a reference replaces the text box for the reference structure ID with a pointer selector for the new reference structure. (**d**) Selecting the custom option for chain matching brings up the custom chain matching dictionary generation wizards. The first wizard (circled in red) enables the replacement of entries in the dictionary by specifying a position and match order in the two boxes with red labels, as shown by the two red arrows. The second wizard (circled in black) recovers items from the dictionary to make it easier to check and update them. It fills the chain order and label boxes, as indicated by the black arrows. (**e**) The full custom match dictionary used in the analysis of the 24 D614G structures is shown, matching our previous study [65]. The first D614G structure determined from intact viruses (used as a reference here) is given chain order BAC and the others have orders BCA and ABC to match it. The rest of the structures have similar chain orders, giving two blocks with an exception for one of the structures (PDB: 7EB4). (**f**) The output of the ensemble building protocol for the 24 D614G structures is shown, highlighting the different objects generated and the summary information provided.

**Figure 5 ijms-24-14245-f005:**
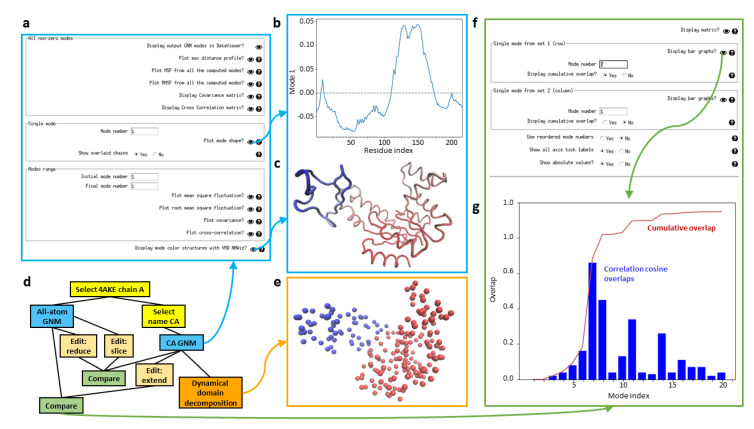
Illustration of GNM and mode comparison viewers. (**a**) The GNM viewer form is shown for the Cα atom GNM from the workflow in panel (**d**), based on an open adenylate kinase structure (PDB: 4AKE, chain A; blue arrow going up). Certain results are highlighted by arrows and shown in panels (**b**,**c**). (**b**) The mode shape is plotted as a line with atoms moving in opposite directions have positive and negative signs. (**c**) The NMWiz viewer colours the structure by the mode shape. (**d**) An example workflow using GNM is shown using open adenylate kinase. Two different atoms selections in yellow precede two GNM calculations in blue using all atoms of chain A and only the Cα atoms. The latter is used for dynamical domain decomposition (orange), and the two of them are used to illustrate mode editing (light orange) and comparison (green). (**e**) The output from the dynamical domain decomposition wizard using two modes (including the zero mode) is a structure coloured by dynamical domains in line with the behaviour in the first non-zero mode shown above. (**f**) The mode comparison viewer is shown for the comparison between the all-atom GNM and the extended CA GNM. (**g**) The result from selecting the option to display the row from comparing mode 7 (non-zero mode 1) from set 1 to all the others with cumulative overlaps enabled is shown. Correlation cosine overlaps are blue bars and the cumulative overlap is shown with a dark red line.

**Table 1 ijms-24-14245-t001:** Challenges in Cryo-EM continuous heterogeneity analysis and potential solutions assisted by Scipion-EM-ProDy.

Problem	Solution	Existing Tools	Scipion-EM-ProDy ^1^
Most analyses of continuous heterogeneity lack biological interpretability and physical meaning (often separate images and maps by non-structural factors)	- Generate representative maps and atomic structures rather than just using landscapes and conformational changes directly from continuous heterogeneity analyses of images - Clustering and dimensionality reduction in structural space	- StructMap for map landscapes based on NMA ^2^, correlations ^3^ or Zernike3D ^3^ - Atomic structure dimensionality reduction, NMA and VMD animation ^2^ - Theoretical motion statistics from NMA of single structures ^2^ or Zernike3D analysis of images ^3^ - Cluster and dimensionality reduce images as structures based on NMA ^2^ or Zernike3D ^3^ - CryoDRGN MAVEn tools ^4^	- PCA, NMA, and deform vectors - NMWiz viewer and comparison tools - GNM and domain decomposition
Large numbers of lower quality maps from continuous heterogeneity are challenging to fit with good structures	- Fix rough fits with short simulations, perhaps w/NMA	- GENESIS NMMD and MD ^3^	- ClustENM(D) for OpenMM, with or without NMA generations
Need to compare standard cryo-EM and continuous heterogeneity outputs to existing structures and those from simulations, and make sense of results in broader context	- Compare cluster representatives from each approach - Compare images directly to projections from maps simulated from structures in a careful way that accounts for prior expectations, such as structural similarity	- BioEM, cryo-BIFE and ensemble reweighting ^5^ - Deformations and landscapes priors ^3^	- Ensemble and trajectory analysis - Atomic structure clustering - Combined PCA

^1^ Scipion-EM-ProDy v3.3.0 (this work); ^2^ ContinuousFlex v3.4.0; ^3^ FlexUtils v3.0.2 and Xmipp v3.23.07; ^4^ MAVEn v1.0; ^5^ Cryo-BIFE and ensemble reweighting code is under development at https://github.com/flatironinstitute/Ensemble-reweighting-using-Cryo-EM-particles (accessed on 6 August 2023)—all outside Scipion.

**Table 2 ijms-24-14245-t002:** Table relating Scipion and ProDy objects.

Scipion Object	ProDy Objects
AtomStruct, SetOfAtomStructs	Atomic ^1^, AtomGroup, Selection
NormalMode	VectorBase ^1^, Vector, Mode
SetOfNormalModes	NMA ^1^, ANM, RTB, GNM
SetOfPrincipalComponents ^2^	PCA
TrajFrame ^3^	Conformation, Frame
SetOfTrajFrames ^3^	Ensemble, Trajectory
ProDyNpzEnsemble ^3^	PDBEnsemble

^1^ Base/parent class; ^2^ new class based on SetOfNormalModes; ^3^ part of Scipion-EM-ProDy at the time of writing.

## Data Availability

Scipion and ProDy are open source software and all code is available on Github at https://github.com/prody/ProDy (accessed on 7 August 2023), https://github.com/scipion-em/scipion-em (accessed on 1 August 2023) and https://github.com/scipion-em/scipion-em-prody (accessed on 6 August 2023). Stable master versions are also available on the Python package index (PyPI) and can be installed with pip and the Scipion plugin manager.

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
