# Peer review of "Scipion-EM-ProDy: A Graphical Interface for the ProDy Python Package within the Scipion Workflow Engine Enabling Integration of Databases, Simulations and Cryo-Electron Microscopy Image Processing"

_ijms, 2023, doi:10.3390/ijms241814245_

Round 1

Reviewer 1 Report

The authors provided an interesting manuscript on the actual topic of macromolecular dynamics.  However, in its current state, the manuscript looks a little messy and is difficult to follow. Everything is mixed with literature analysis, and the authors' achievements become unclear.  The manuscript does not correspond to the typical structure; the results chapter goes directly after the introduction. Material and methods come after discussion. I believe it is required to restructure the text, clearly separating state-of-the-art and research sections. Authors' contributions to the research field must be clarified, and achievements should be highlighted.

 Minor remarks:

In line 42, the reference is faulty. In line 373 faulty reference to the subchapter.

The titles of subchapters (1.1, 1.2) sound as independent statements. I recommend modifying them.

The titles of the figures are incredibly long. I recommend moving a significant part into the main text.

English quality is fine.

Author Response

The authors provided an interesting manuscript on the actual topic of macromolecular dynamics. 

Thank you

However, in its current state, the manuscript looks a little messy and is difficult to follow. Everything is mixed with literature analysis, and the authors' achievements become unclear.  The manuscript does not correspond to the typical structure; the results chapter goes directly after the introduction. Material and methods come after discussion. I believe it is required to restructure the text, clearly separating state-of-the-art and research sections. Authors' contributions to the research field must be clarified, and achievements should be highlighted.

We thank the reviewer for these helpful comments and have done the following to make our achievements clearer:

  1. We have rearranged Table 1, moving our general Scipion Flexibility Hub tools from our previous paper into the column of existing tools
  2. We have made several clarifying changes to the Introduction, including to subheading 1.1 to highlight “Recent solutions” in general rather than our previous Scipion Flexibility Hub in particular
  3. We have changed the layout and moved the methods section up as requested. We have also moved the implementation overview and technical details on different chain orders from the results to the methods. We are sorry that the chosen layout was confusing and hope this is much clearer.

Minor remarks:

In line 42, the reference is faulty. In line 373 faulty reference to the subchapter.

Fixed. Thank you

The titles of subchapters (1.1, 1.2) sound as independent statements. I recommend modifying them.

We have now adjusted these. Thanks

The titles of the figures are incredibly long. I recommend moving a significant part into the main text.

We have adjusted some of these. Thanks

Reviewer 2 Report

This article details the incorporation of the ProDy Python package into the Scipion workflow engine. While ProDy is a renowned tool for analyzing global protein dynamics using normal mode analysis (NMA) and principal component analysis (PCA), it's predominantly accessible to seasoned programmers. By integrating ProDy with Scipion, a broader user base can now leverage its features via a graphical interface. The derived plugin, Scipion-EM-ProDy, not only streamlines the use of ProDy but also seamlessly fits into expansive workflows catered to cryo-electron microscopy image analysis and molecular simulations. The comprehensive capabilities of Scipion-EM-ProDy encompass atomic structure manipulations, ensemble analysis, and hybrid simulations, further simplifying the interpretation and simulation of atomic structures through swift computational biophysics. I recommend acceptance without modifications.

Author Response

We thank Reviewer 2 very much for their extremely positive comments.

We did, however, have to make considerable changes to the manuscript in response to Reviewer 1 (see below) and hope that Reviewer 2 is still happy with it.

The authors provided an interesting manuscript on the actual topic of macromolecular dynamics. 

Thank you

However, in its current state, the manuscript looks a little messy and is difficult to follow. Everything is mixed with literature analysis, and the authors' achievements become unclear.  The manuscript does not correspond to the typical structure; the results chapter goes directly after the introduction. Material and methods come after discussion. I believe it is required to restructure the text, clearly separating state-of-the-art and research sections. Authors' contributions to the research field must be clarified, and achievements should be highlighted.

We thank the reviewer for these helpful comments and have done the following to make our achievements clearer:

  1. We have rearranged Table 1, moving our general Scipion Flexibility Hub tools from our previous paper into the column of existing tools
  2. We have made several clarifying changes to the Introduction, including to subheading 1.1 to highlight “Recent solutions” in general rather than our previous Scipion Flexibility Hub in particular
  3. We have changed the layout and moved the methods section up as requested. We have also moved the implementation overview and technical details on different chain orders from the results to the methods. We are sorry that the chosen layout was confusing and hope this is much clearer.

Minor remarks:

In line 42, the reference is faulty. In line 373 faulty reference to the subchapter.

Fixed. Thank you

The titles of subchapters (1.1, 1.2) sound as independent statements. I recommend modifying them. We have now adjusted these. Thanks

The titles of the figures are incredibly long. I recommend moving a significant part into the main text. We have adjusted some of these. Thanks

Round 2

Reviewer 1 Report

Authors take into acount major remarks and improved the structure of their article. In my opinion now it is suitable for publication.

The quality of the language is fine.